# Amorphous Inclusion Complexes: Molecular Interactions of Hesperidin and Hesperetin with HP-Β-CD and Their Biological Effects

**DOI:** 10.3390/ijms23074000

**Published:** 2022-04-04

**Authors:** Kamil Wdowiak, Natalia Rosiak, Ewa Tykarska, Marcin Żarowski, Anita Płazińska, Wojciech Płaziński, Judyta Cielecka-Piontek

**Affiliations:** 1Department of Pharmacognosy, Poznan University of Medical Sciences, Rokietnicka 3, 60-806 Poznan, Poland; kamil.wdowiak@student.ump.edu.pl (K.W.); nrosiak@ump.edu.pl (N.R.); 2Department of Chemical Technology of Drugs, Poznan University of Medical Sciences, Grunwaldzka 6, 60-780 Poznan, Poland; etykarsk@ump.edu.pl; 3Department of Developmental Neurology, Poznan University of Medical Sciences, Przybyszewski 49, 60-355 Poznan, Poland; zarowski@ump.edu.pl; 4Department of Biopharmacy, Faculty of Pharmacy, Medical University of Lublin, Chodzki 4a, 20-093 Lublin, Poland; anita.plazinska@umlub.pl; 5Jerzy Haber Institute of Catalysis and Surface Chemistry, Polish Academy of Sciences, Niezapominajek 8, 30-239 Krakow, Poland; wojtek_plazinski@o2.pl

**Keywords:** hesperidin, hesperetin, hydroxypropyl-β-cyclodextrin, solubility, amorphous inclusion complex

## Abstract

This study aimed at obtaining hesperidin (Hed) and hesperetin (Het) systems with HP-β-CD by means of the solvent evaporation method. The produced systems were identified using infrared spectroscopy (FT-IR), X-ray powder diffraction (XRPD), and differential scanning calorimetry (DSC). Moreover, in silico docking and molecular dynamics studies were performed to assess the most preferable site of interactions between tested compounds and HP-β-CD. The changes of physicochemical properties (solubility, dissolution rate, and permeability) were determined chromatographically. The impact of modification on biological activity was tested in an antioxidant study as well as with regards to inhibition of enzymes important in pathogenesis of neurodegenerative diseases. The results indicated improvement in solubility over 1000 and 2000 times for Hed and Het, respectively. Permeability studies revealed that Hed has difficulties in crossing biological membranes, in contrast with Het, which can be considered to be well absorbed. The improved physicochemical properties influenced the biological activity in a positive manner by the increase in inhibitory activity on the DPPH radical and cholinoesterases. To conclude the use of HP-β-CD as a carrier in the formation of an amorphous inclusion complex seems to be a promising approach to improve the biological activity and bioavailability of Hed and Het.

## 1. Introduction

Flavonoids are bioactive phytochemical compounds that have a wide range of bioactivity in the terms of pro-healthy effect on human health [1,2]. They can be present in two forms: (i) glycoside form, characterized by the presence of sugar moiety attached to typical flavonoid structure, the so-called aglycone part, and (ii) the aglycone form itself [3]. Glycoside forms, as they show relatively greater solubility than aglycones, play an essential role in plant physiology [4]. However, as far as potential health benefits are concerned, the crucial form seems to be the aglycone form.

Hesperidin (Hed) and its aglicone hesperetin (Het) are both flavanones that can be found in citrus fruits such as sweet oranges, clementines, mandarins, lime, lemon, and grapefruit [5]. They possess a broad spectrum of pharmacological activity, including anticancer [6], antidiabetic [7], neuroprotective [8], and antimicrobial [9] properties. They also show a protective effect on the cardiovascular system [10]. To highlight the pro-healthy potential of Hed and Het, it is worth mentioning that these flavonoids seem to be solutions for the on-going global pandemic caused by severe acute respiratory syndrome coronavirus 2 (SARS-CoV-2). There are some papers suggesting the effectiveness and mechanism of action on virus infection. The proposed modes of action include suppression of entrance of the virus into the cell via hindering the binding of angiotensin converting enzyme-2 receptors with viral spike protein, and inhibition of proteases, which are essential for transformation of early proteins of the virus to enable further viral replication [5,11,12].

In gastrointestinal tract conditions, the glycoside form undergoes a transformation, triggered by the intestinal microbiome and its enzymes, to release the aglycone form, which is absorbed and reaches systemic circulation [13]. What is more, flavonoids as well as the detached sugar part can play the role of substrate, stimulating the growth of probiotic bacteria and therefore having a meaningful impact on microflora [13,14,15]. Both glycoside and aglycone forms have immense potential in treatment of chronic diseases.

The use of polyphenol compounds is limited by poor bioavailability. One of the main factors is meager aqueous solubility, which leads to limited transmembrane permeability [16]. Conversely, cyclodextrins are recognized as effective solubilizators.

Cyclodextrins are known to form inclusion complexes due to possessing hydrophobic cavity, to which poorly soluble, lipophilic compounds have a high affinity. One the other hand, hydroxyl groups of the CD are oriented on the outer surface of CD molecule, making it hydrophilic. The dual nature of CD is responsible for its ability to increase solubility of poorly soluble drugs [17,18,19]. Apart from creating inclusion complexes, CD are said to form non-inclusion complexes, which are related to self-assembling of CD and forming aggregates [19,20,21,22]. Therefore, the improvement of solubility can be related to different mechanisms.

In the case of Hed, some studies can be found in which it was combined with various CDs. Corciova et al. reported that complexing Hed with β-CD and HP-β-CD resulted in improved dissolution as well as antibacterial, antioxidant, and lipoxygenase inhibitory activities. They also focused on calculating thermodynamic parameters of complexation reaction, revealing that this process in this case is spontaneous [23,24]. When it comes to coupling Het with CD, Sangpheak et al. produced its complexes with β-CD and methylated derivatives. The study provided evidence that mentioned CDs enabled enhancement in the terms of dissolution rate as well as anti-inflammatory and cytotoxicity effects towards cancer cell lines [25]. Lucas–Abellan et al. prepared Het complexes with β- and HP-β-CD, which was linked to improved solubility by 30- and 467-fold, respectively. They also indicated that pH value has an impact on complexation, wherein the strongest complexes of Het were formed at pH 6.5, at which Het has the protonated form [26]. So far, cyclodextrins have helped to improve the solubility of several polyphenols such as caffeic acid [27], ellagic acid [28], quercetin [29,30], rutin [30], resveratrol [31], and myricetin [1]. One of the best studied flavonoids in this term is rutin (quercetin-3-O-rutinoside). In the literature, there are reports suggesting the improvement of its solubility by forming inclusion complexes with cyclodextrins (CDs) [32,33]. Moreover, there are also reports concerning the enhancement of solubility of quercetin and rutin with CDs [29,30]. In the case of these compounds, the most energetically favored interactions arise between the inner cavity of β-CD and dihydroxyphenyl and dihydroxychromone groups [33].

Our studies involved (i) the preparation and identification of Hed and Het systems with HP-β-CD based also on in silico study; (ii) studies of physicochemical properties in the terms of solubility, dissolution rate, and permeability; and (iii) assessment of the impact on biological activity of inclusion complexes with regards to antioxidant potential as well as acetylcholinesterase and butyrylcholinoesterase inhibition.

## 2. Results

### 2.1. Preparation of Systems of Hesperidin and Hesperetin with HP-β-CD

Systems of Hed and Het with HP-β-CD were prepared by the means of solvent evaporation method. The systems looked like a slightly yellowish glistening fine powder. Chromatographic analysis indicated that the preparation method of the systems was safe, meaning that the used method did not cause sugar moiety detachment when it comes to the chemical stability of Hed. Chromatograms of Hed and Het for the developed method are found in the Appendix A.

### 2.2. Identification of Hesperidin and Hesperetin with HP-β-CD Systems

#### 2.2.1. Fourier Transform Infrared Spectroscopy

The infrared spectra of the Hed/HP-β-CD systems (ratio 1:1 and 1:2) were analyzed and compared with the spectra of the pure compounds and their physical mixtures (PM), respectively (see Figure 1).

The most characteristic bands for Hed and HP-β-CD and their assignments are listed in Appendix A. The FT-IR spectrum of the physical mixture is the superposition of two compounds. However, the spectra of Hed/HP-β-CD systems showed no features similar to pure Hed. For example, peaks observed in the Hed spectrum in the range of 400–1000 cm^−1^ (O–H, C–H, C–O–C, and C–C vibration at glucose, rhamnose, and A ring) disappeared completely in systems, whereas the bands characteristic for HP-β-CD were still visible (see the Figure 1), including peaks at 847 cm^−1^ and 947 cm^−1^ corresponding to the presence of glucopyranose units of HP-β-CD in C1 chair conformation [34]. This suggests that A ring and rutinose moiety are located in the proximity of the hydroxymethyl groups and ring oxygen atoms of cyclodextrin. The inclusion of a guest molecule in the CD cavity did not affect the vibrational bands of the C–O–C groups observed in the 1000–1200 cm^−1^ range for HP-β-CD. After complexation, the changes were visible among others for the bands 1277 cm^−1^ (C–O–C and O–H at B ring and C–H at C ring), 1605 cm^−1^ and 1645 cm^−1^ (C=O at C ring), and 1441 cm^−1^ and 1518 cm^−1^ (C–H in methoxy group at B ring and C–H at C ring, respectively). The intensities of these peaks were reduced drastically and/or broadened considerably. Conversely, the band observed in Hed at 1441 cm^−1^ in the systems disappeared completely. Changes in stretching vibrations of the carbonyl group in Hed/HP-β-CD systems suggest a change in the environment of the C=O group at the C ring of Hed [35]. The drastic decrease of the intensities of characteristic bands of Hed suggests the incorporation of B and C ring into the host cavity [36]. Next, disappearance in the 2850–3100 cm^−1^ spectral region band corresponding to the C–H stretching vibration of Hed and the shift of the wide band at about 3300 cm^−1^ corresponding to the O–H group (in HP-β-CD and Hed, see the Appendix A) confirmed the formation of hydrogen bonds between Hed and HP-β-CD.

The infrared spectra of the Het/HP-β-CD systems (ratio 1:1 and 1:2) were analyzed and compared with the spectra of the pure compounds and their physical mixtures, respectively (see Figure 2).

The most characteristic bands for Het and HP-β-CD are compared with the assignments in Appendix A. The FT-IR spectrum of the physical mixture is the superposition of two compounds. However, the spectra of the Het/HP-β-CD systems showed no features similar to pure Het. For example, bands corresponding to the vibration at B and C ring observed in Het at 878, 957, 1024, 1063, and 1092 cm^−1^ in spectra of the Het/HP-β-CD systems completely disappeared. As for Hed, this suggests the incorporation of B and C rings into the host cavity [36].

Then, the differences in the 1200–1700 cm^−1^ spectral region were analyzed in terms of the implication of different molecular groups of the guest and the host molecules in the inclusion process [36]. In this range, predominant bands corresponded to the C–H and O–H vibrations at B ring, C–H and C=O vibrations at C ring, and C–H vibration of HP-β-CD (see Appendix A). In the FT-IR spectra of Het/HP-β-CD systems, most of the characteristic Het bands disappeared; the bands at 1202, 1337, 1503, 1576, and 1634 cm^−1^ were reduced in intensity and shifted, while the bands corresponding to the C–H group observed in the HP-β-CD spectrum were shifted. Additionally, disappearance in the 2840–3000 cm^−1^ spectral region band corresponding to the C–H stretching vibration of B and C ring of Het and the shift of the wide band at about 3300 cm^−1^ corresponding to the O–H group in HP-β-CD (see Appendix A) was observed. These changes confirm the formation of hydrogen bonds between Het and HP-β-CD.

Ji-Sang Kim [37] indicated that the fundamental changes that appear in the FT-IR spectra of inclusion complexes of flavonoid with HP-β-CD are reflected mainly in the C=O stretching spectral region. These observations indicate the presence of a similar complexing mechanism for the guest molecule (flavonoid) in the host cavity (cyclodextrin molecule), which is reflected in our results.

#### 2.2.2. X-ray Powder Diffraction

X-ray powder diffraction is a useful method for identifying crystalline and amorphous phases.

Figure 3 presents XRPD diffractograms of pure Hed, Het, and HP-β-CD components; their physical mixtures; and their binary systems in molar ratios 1:1 and 1:2 (API to HP-β-CD). Both flavonoid diffraction patterns revealed sharp peaks characteristic of crystalline materials. Peaks with relative intensity above 40% were detected at diffraction angles (2Ɵ) of 12.16°, 15.55°, 19.59°, 21.26°, 22.41°, and 24.80° for Hed and at 14.53°, 16.94°, 26.23°, and 29.46° for Het. Appendix A shows the full list of peaks, their relative intensities, Bragg angles 2Ɵ, and interplanar distances d_hkl_. Comparison of Het diffractogram with those generated from monohydrate [38] and anhydrous [39] hesperetin single-crystal structures deposited in the Cambridge Structural Database (CSD) [40] revealed that the powder pattern of Het corresponds to an anhydrous crystal form.

The broad halo pattern of HP-β-CD confirmed its amorphous nature. The XRPD patterns of the physical mixtures of Hed and Het with HP-β-CD in molar ratios 1:1 and 1:2 corresponded to the superposition of the patterns of individual components, indicating no new solid-state phase formation. In contrast, their binary systems exhibited the diffused diffraction patterns indicative of the complete amorphization of Hed and Het in these products. The results can be attributed to intermolecular interactions between components of the system and possible complex formation, as indicated by FT-IR spectroscopy and in silico docking.

#### 2.2.3. Differential Scanning Calorimetry

Differential scanning calorimetry (DSC) was used to recognize the thermal properties of tested compounds. Each measurement involved two heating cycles. In the first cycle, the samples were heated from 25 °C to 100 °C in order to remove any traces of water [41,42,43]. In the second cycle, spanning 25–275 °C, the thermograms shown in Figure 4 were recorded.

The DSC curves of flavonoids showed sharp endothermic peaks at 259 °C and 234 °C attributed to the melting points of Hed and Het, respectively [43]. The melting endotherm disappeared in scans of 1:1 and 1:2 flavonoid–cyclodextrin binary formulations, which indicated loss of crystallinity Hed and Het, as well as formation intermolecular interactions between the system’s two components. According to the literature [37,44,45], when the guest molecule is embedded in the cyclodextrin cavity to form an amorphous inclusion complex, its melting point is either shifted to a different temperature or disappears. Therefore, the obtained results supplement FT-IR analysis.

### 2.3. Physicochemical Properties

#### 2.3.1. Solubility Studies

Solubility studies were performed in three different media, namely, were water, phosphate buffer (pH 6.8), and HCl 0.1 N. Concentrations of Hed and Het were determined by chromatography. Validation parameters are presented in the Appendix A. The results of the assay are presented in Table 1. The solubility of the tested pure compounds was far below 0.1 mg·mL^−1^, classifying these compounds as a practically insoluble [46]. As presented, the addition of HP-β-CD led to an increase of solubility of polyphenols. In both cases, the best solubility enhancement was provided by the obtained systems in a 1:2 molar ratio in all of the media. Interestingly, the addition of HP-β-CD itself, on the basis of the results of the physical mixtures’ solubility results, caused the increase in solubility, so the solubilizing effect of CD was evident and provided a remarkable improvement in solubility.

As far as an impact of pH conditions on solubility are concerned, it can be observed that in a more basic environment (phosphate buffer pH 6.8), tested compounds are more soluble than in acidic media. This can be related to weak acid character of flavanones and the fact of dissociation in basic medium [26,47].

In previous studies, the Higuchi solubility diagrams were calculated to characterize formation of inclusion complexes between Hed and Het with HP-β-CD in aqueous solution. Studies revealed that diagrams can be described as AL type, meaning that there is an increase of solubility in linear host–guest correlation [23,48].

#### 2.3.2. Dissolution Rate Studies

Complexing tested polyphenols with HP-β-CD had a remarkable impact on dissolution behavior. The impact on dissolution rate was evident, taking into account that the obtained systems provided incredibly fast dissolution that translated into reaching a “plateau” state just after 15 min. The increase in apparent solubility was also noticeable. It is worth noting that complexes, in the case of 1:2 molar ratio systems, enabled dissolving of an almost total amount of compounds. The results of dissolution rate studies are in line with our findings in solubility studies since systems providing the best enhancement in solubility also presented the highest apparent solubility. The results of dissolution rate studies are presented in Figure 5a,b.

#### 2.3.3. Permeability Studies

To assess the impact of obtained systems on permeability of tested compounds, the PAMPA assay (the Figure 6a,b), simulating the passive diffusion mode of absorption through gastrointestinal walls, was applied. To determinate P_app_ factor, we dissolved the pure compounds in DMSO, strictly following assay guidelines. Further, we also prepared samples by dissolving systems in water to check if the solubility enhancement of Hed and Het, had an influence on passive absorption of the tested compounds. The GIT PAMPA assay revealed that Hed was poorly absorbed. Moreover, the obtained systems did not affect permeability by enhancing the passive transport across the membrane. On the other side, Het and its systems, from calculated factors, can be considered to be readily absorbed in the intestines.

The BBB PAMPA assay is meant to simulate the blood–brain barrier; therefore, it enables the determination as to whether a compound can enter the central nervous system. The results of this assay are similar to those of GIT assay, meaning that the Hed and its systems can have difficulties to cross the blood–brain barrier, whereas Het and its complexes would be easily able to pass this biological barrier.

### 2.4. Biological Assays

#### 2.4.1. Antioxidant Activity Assay

In the DPPH antioxidant assay (the Figure 7), there was a correlation between improved solubility and enhanced antioxidant activity. The best DPPH radical inhibitory activity, i.e., 82.5 ± 0.84% in the case of Hed systems was Hed/HP-β-CD in a 1:2 molar ratio, whereas when looking at Het systems, it was Het/HP-β-CD in a 1:2 molar ratio by inhibiting 93.48 ± 0.47% of the radical. In sum, both of these systems provided the best solubility enhancement in the case of each compound.

#### 2.4.2. Acetylcholinesterase (AChE) and Butyrylcholinesterase (BChE) Inhibition Assay

As in the antioxidant assay, the AChE and BChE inhibition assays (Figure 8a,b) provided evidence on the influence of improved solubility in biological activity in a positive manner, at least in in vitro studies. In the AChE study, the best inhibitory activity was presented for the system of the Hed/HP-β-CD 1:2 molar ratio and the system of the Het/HP-β-CD 1:2 molar ratio by inhibiting 55.6 ± 1.60% and 56.04 ± 2.91%, respectively. Similar findings were reported in the BChE assay, wherein the same systems inhibited the mentioned enzyme at 67.24 ± 2.87% and 77.54 ± 0.93% for Hed/HP-β-CD 1:2 and Het/HP-β-CD 1:2 systems, respectively.

### 2.5. Docking

All types of low-energy complexes identified during the docking procedure exhibited fairly similar orientations of the guest molecules in the binding cavity of cyclodextrins. Namely, we obtained a series of expected guest–host complexes, with guest molecules located in the center of the binding cavity formed by the inner channel of the cyclodextrin molecule. Observations given in this subsection relied on inspection of multiple binding poses recovered during the docking study and characterized by only minor relative differences in binding energies (≈1–≈4 kJ/mol, depending on the system) (see Table 2).

Both studied compounds contained the same topological pattern in their molecular structure (i.e., the 3′,5-dihydroxy-4′-methoxy moiety) that facilitates the comparisons of their binding poses (see Figure 9). This fragment of guest molecules containing both aromatic and saturated rings has been identified as an anchoring region responsible for favorable ligand–CD interactions.

The orientation of the guest molecule in the binding cavity was always the same, i.e., the double-ring moiety was directed towards that edge of the 2HP-β-CD torus, which contained only the hydroxyl (or hydroxyl and 2HP*) groups. The opposite, single-ring moiety was located closer to the hydroxymethyl groups and ring oxygen atoms. At the same time, the disaccharide group of hesperidin exhibited more (2HP-β-CD*) or less (2HP-β-CD^#^) intensive contacts with 2HP groups of 2HP-β-CD hosts, depending on the substitution type. Such a general pattern of the guest–host interactions was fairly well maintained across the complete set of studied guest–host pairs; however, a number of minor differences between the observed contacts was identified as well. Differences mainly involved the most flexible groups of guest molecules, i.e., 2HP, hydroxymethyl, and hydroxyl moieties. The interactions with such groups (involving intermolecular hydrogen bonding) were not maintained across the whole set of low-energy structures. A relatively low scatter in determined binding energies, accompanied by notable structural differences, suggested the high flexibility of the whole molecular complexes, independently of the considered substitution pattern.

The calculated binding energies varied between ≈−20–≈−30 kJ/mol (see Table 2), indicating strongly favorable interactions between guest and host molecules. The magnitude of binding energies seemed to be correlated with the size of interacting molecules.

### 2.6. Molecular Dynamics

The structural topology of the guest–host complexes obtained during docking as well as the most essential intermolecular contacts were maintained during MD simulations. More precisely, the double-ring moiety is still directed towards that edge of the host molecule that contained only the hydroxyl (or hydroxyl and 2HP*) groups. The opposite, single-ring moiety was located in the proximity of the hydroxymethyl groups and ring oxygen atoms of cyclodextrin. In all cases, the most characteristic and conserved contact type was that delivered by the aromatic moieties of the guest molecules interacting with the inner part of the cyclodextrin cavity. The chemical nature of the interacting molecular fragments suggests that such contact is possible due to the existence of the CH–π attractive interactions. Note that although modern molecular mechanics force fields do not contain the explicit functional forms responsible for reproducing such interactions, the combination of non-bonded (Lennard–Jones and Coulombic) interactions is fully capable of effectively mimicking them [49]. Therefore, such a mechanism of binding can be investigated by using the accepted methodology. As indicated by systematic changes in the SASA values (Table 2), the solvent-exclusion effects can also be significant.

Figure 9 contains the radial distribution functions (RDFs) illustrating the most essential contacts present in the studied guest–host complexes. The position of the two rings was only slightly altered in comparison to the structures resulting from the docking studies. Namely, the anchoring contacts were delivered by the common topological motif of both guest molecules, i.e., the 3′,5-dihydroxy-4′-methoxy moiety. In the case of 2HP-β-CD hosts, none of the two alternative substitution patterns were correlated with any substantial differences in binding mechanism or structural rearrangements within the complex core. As can be seen in Figure 8, the course of RDFs was very similar, independently of the part of the 3′,5-dihydroxy-4′-methoxy moiety, indicating their similar magnitude of intermolecular contacts. However, taking into account the larger size of the double-ring structure and slightly broader peaks corresponding to this fragment, it can be concluded that this type of interaction is slightly weaker in comparison to those delivered by a single-ring fragment. Thus, the substituted phenyl ring can be ranked as the most substantial in binding of both guest molecules to cyclodextrins. The presence of a large disaccharide moiety in the case of the hesperidin molecule did not affect the binding pattern, in analogy to the presence of 2HP substituents in the host molecules. The position of the guest molecule was always roughly fixed with respect to the “core” of cyclodextrin, while remaining substituents were able to interact with each other on opportunistic bases. The exemplary snapshots of the MD trajectory are illustrated in Figure 10.

The large flexibility of the complex assumed on the basis of the docking results was fully confirmed by the significant variation associated with the broadness of RDF peaks (see Figure 8). All the guest–host complexes were not rigid but displayed a high degree of conformational heterogeneity, including both the guest molecules (reorientations around the –C–C– and –C–O– rotatable bonds were observed) and the whole complex. A visual inspection of the MD trajectory revealed that the identified contacts could involve various chemically identical carbohydrate residues across the cyclodextrin perimeter.

The analysis of structural features of guest–host complexes was complemented by the inspection of energies involved in stabilizing them. Table 2 contains a short summary of such an analysis. In spite of configurational feasibility (i.e., the high proximity of the 2HP, hydroxymethyl, and hydroxyl groups of host molecules and the hydroxyl, carbonyl, and methoxy groups of guests), hydrogen bonding between guest and host molecules were relatively scarce and varied between ≈0.6 and ≈2 occurrences per timeframe, depending on the system. This can be explained by the influence of water molecules surrounding the guest–host complexes and saturating the hydrogen bond donors and acceptors present in the complex. Moreover, it emphasizes the importance of other, non-polar interactions. The most frequent hydrogen bonding corresponded to the hesperidin+2HP-β-CD^#^ complex, which was the result of a close proximity of 2HP groups and polar substituents on the phenyl ring of the guest.

The calculated energies of interactions between guest and host molecules cannot be directly associated with the binding free energies (and, thus, guest–host affinities) but they still clearly indicated a strong, favorable association. The magnitude of energies corresponding to short-range, non-bonded interactions between guest and host molecules varied between −107 kJ/mol and −184 kJ/mol. A relatively minor contribution of the electrostatic component (varying between 12 and 29%) suggested that the binding process is driven through solvent effects (e.g., minimizing the area of non-polar surface) or other non-polar interactions (e.g., CH–π interactions, mentioned above).

The calculations of the changes in the average values of the SASA occurring upon complexation were performed. The values of this parameter were quite similar across the studied systems (≈4.2–6.5 nm^2^), and the presence of 2HP groups was always correlated with its increase. This spoke for a favorable role of this type of substitution in the context of guest–host affinity. Moreover, similar changes in the SASA values spoke for roughly equally favorable spatial accommodation of the guest molecules in the host binding cavity.

The above analysis, considering the two alternative substitution patterns of the 2HP groups, had no aim in identifying the most favorable binding mode and to discriminate between those two possibilities. As the real substitution pattern is unknown, any of these above two possibilities can occur, as well as their combination or yet another substitution pattern (e.g., an intermediate between the limiting cases discussed here). A series of similarities between binding parameters discussed here and determined for different 2HP-β-CDs allows for the assumption that the obtained binding characteristics is a fair representation for some more diverse systems containing 2HP substituents.

Finally, let us mention that the determined binding characteristics was extremely similar to that investigated by us in our previous work [50] and concerning naringenin bound by 2HP-β-CD. This is fully understandable as either naringenin or the presently studied compounds contain the same topological motif in their molecular structure. Thus, it can be concluded that the binding patterns described here and in [50] are characteristic of a larger group of compounds sharing the same molecular moiety.

## 3. Discussion

Both Hed and its aglycone Het have a vast pro-healthy mode of action and could therefore be used as prophylactic agents or adjuvants in current treatment. However, their low bioavailability leads to insufficient blood concentrations for instigating the desired therapeutic effect, thus limiting their true pharmacological potential. The increase in solubility can offer the solution to this problem and may turn out to be a hope for improvement of the bioavailability of the studied compounds.

In this paper, our approach was to obtain systems of Hed and Het with HP-β-CD to enhance physicochemical properties and, consequently, their biological activity.

The solvent evaporation technique was the method that enabled the formation of Hed or Het complexes with HP-β-CD. The systems were created in molar ratios of 1:1 and 1:2 with an increasing concentration of cyclodextrin. We focused on providing evidence that obtaining amorphous inclusion complexes has an impact on solubility, dissolution rate, and permeability, which further can be translated into better neuroprotective activity.

FT-IR studies confirmed the applied solvent evaporation technique allowed to obtain the inclusion complexes, while the most preferable site of interactions between Hes or Hed with HP-β-CD was studied by using the in silico docking and molecular dynamics. Given that the FT-IR studies and in silico studies indicate the formation of an inclusion complex, it seems likely that the changes observed in the XRPD pattern (absence of noticeable crystalline peaks) and in the DSC thermogram (lack of endothermic peak corresponding to melting point of tested compounds) resulted from the formation of the complex.

In the FT-IR spectra of Hed/HP-β-CD and Het/HP-β-CD systems, most of the characteristic Hed and Het bands corresponding to the B and C rings disappeared, reduced in intensity, and/or were shifted. Moreover, the bands corresponding to the C–H group observed in the HP-β-CD spectrum were shifted in all systems. The drastic changes of characteristic bands of Hed and Het suggested the incorporation of B and C rings into the host cavity. Since the C–O–C groups existed on the surface of the bucket-like structure of cyclodextrin, the inclusion of Hed or Het guest molecules in the HP-β-CD cavity did not affect this vibrational mode (range 1000–1200 cm^−1^). A shift of the wide band at about 3300 cm^−1^ corresponding to the O–H group (in HP-β-CD and Hed) further confirmed the existence of hydrogen bonds between Hed or Het and HP-β-CD.

In his work, Srirangam suggested the formation of the hesperidin/HP-β-CD complex based on changes in only one band (1644 cm^−1^, carbonyl stretching vibration) [51]. In turn, Corciova et al. [23] attributed differences in FT-IR spectra of hesperidin, HP-β-CD, and their systems to the formation of an inclusion complex between these two components; however, the hesperidin/HP-β-CD spectrum has numerous bands characteristic of pure hesperidin. In our study, virtually no bands of hesperidin in FT-IR spectra of Hed/HP-β-CD systems suggested the existence of complexes with stronger hydrogen bonding interactions [23].

Molecular dynamics analysis indicated the inclusion of the B and C rings of the Hed and Het molecule inside the HP-β-CD. This was confirmed by the results of the FT-IR analysis. In addition, the substituted phenyl B ring can be ranked as the most substantial in the binding of both guest molecules to HP-β-CD. The chemical nature of interacting molecular fragments suggests that such contact is possible due to the existence of the CH–π attractive interactions. The determined binding characteristics is extremely similar to that investigated in our previous work concerning naringenin bound by 2HP-β-CD [50]. This is fully understandable as both naringenin and the presently studied compounds contain the same topological motif in their molecular structure.

FT-IR and molecular dynamics analysis indicated that hydrophobic forces together with a contribution of hydrogen bonding are responsible for the stability of Hed/HP-β-CD and Het/HP-β-CD systems.

In order to evaluate the influence of obtained systems on solubility, dissolution studies were performed. They revealed that the formation of complexes with HP-β-CD led to a significant improvement in the solubility of flavonoids. The best solubility reaching over 1000-fold in the tested solutions was achieved for Hed complexes with the highest amount of HP-β-CD. In the case of Het, the binary systems also revealed a considerable enhancement of solubility, which exceeded 12.0 mg·mL^−1^ in the tested solutions for systems with a molar ratio of 1:2 Het/HP-β-CD. We can say that we were able to convert our compounds from practically insoluble to sparingly soluble [52]. Since solubility is strongly connected to absorption, we can assume that our approach could be guided to an increased bioavailability. Interestingly, physical mixtures also showed better performance. Such a shift in the solubility equilibrium of pure polyphenols can be related to the interactions of HP-β-CD and Hed/Het dissolved in medium, which lowers the concentration of sparingly soluble substances and leads to an increase in their solubility.

To evaluate the behavior of the obtained systems in intestinal conditions in terms of the ability to maintain supersaturation, the dissolution rate study was performed in phosphate buffer (pH 6.8) for 360 min. The dissolution rate studies revealed that the complexation of the tested compounds with HP-β-CD provided a higher apparent solubility, enabling a higher amount of Hed/Het to dissolve. Moreover, the systems managed to maintain a supersaturation state for the time of the study’s duration. Interestingly, the physical mixtures provided dissolution improvement as well. However, it was not as high as in the case of the amorphous inclusion complexes.

Since CD forms complexes with individual molecules, there is a preventive effect on drug crystallization because of difficulties in the self-assembly of small molecules [53]. The results of our study suggest a significant increase in Hed and Het solubility, while dissolution rate studies are characterized by increased apparent solubility; hence, they prove that the obtained systems generated a supersaturation state of both tested compounds [54,55]. Therefore, the prevention of crystallization is a crucial factor in magnifying the advantageous effects in terms of solubility enhancement.

To provide details on Hed and Het bioavailability with regards to the ability to cross biological barriers, the PAMPA assays simulating intestinal conditions (GIT) and the blood–brain barrier (BBB) were performed. According to PAMPA GIT (P_app_ = 5.54 × 10^−8^ ± 2.21 × 10^−8^ cm·s^−1^) and BBB (P_app_ = 5.58 × 10^−7^ ± 2.69 × 10^−9^ cm·s^−1^) results for compounds dissolved in DMSO, the Hed molecule should be considered poorly absorbed and have difficulty crossing the blood–brain barrier. Permeability studies revealed that Het can be described as the molecule that has a chance to be absorbed in the gastrointestinal tract (PAMPA GIT P_app_ = 9.61 × 10^−6^ ± 2.34 × 10^−8^ cm·s^−1^) and that crosses the blood–brain barrier (PAMPA BBB P_app_ = 5.58 × 10^−6^ ± 2.70 × 10^−8^ cm·s^−1^). It is worth noting that the solvent evaporation method did not cause any harm when it came to the permeability of the compound in obtained systems. The obtained permeation factors indicated that Het is well absorbed, taking into account the PAMPA GIT and BBB assays’ data interpretation instructions. In our study, we assessed the permeation ability of compounds with regards to passive transport, so other possibilities, such as active transport, were not considered in the study. Moreover, the Hed was transformed in the colon into Het by intestinal microflora, which as we managed to reveal was well absorbed.

On the basis of our studies, concerning solubility and permeability, we can classify Hed and Het with regards to the BCS (Biopharmaceutical Classification System). Hed can be classified in BCS IV class since it shows poor solubility and poor permeability. On the other hand, Het could be called a BCS class II compound due to its poor solubility and high permeability [56].

Bearing in mind the mentioned relation, we concluded that our systems could provide an improvement in bioavailability as the increased solubility, dissolution rate, and generated supersaturation were found to be strictly related to permeability and reached appropriate bloodstream concentrations to show therapeutic effect. The are several studies aiming at affecting the bioavailability of tested compounds, mentioned in the literature. Saad et al. managed to produce solid lipid nanoparticles of Hed, which translated into nearly 4.5-fold higher bioavailability in the rat model [25]. Zeng et al. generated the nanoemulsion of Het, which improved oral bioavailability by 5.67-fold in terms of AUC [57]. Gu Su-Fang et al. obtained the D-α-tocopheryl polyethylene glycol 1000 succinate micelles and phosphatidylcholine complexes with Het, which translated into 16.2- and 18.0-fold increased AUC of plasma concentration after oral administration [58].

Alzheimer’s disease seems to be a growing problem, especially in the aging population. Cholinesterases are considered to be one of the important molecular targets in the pathophysiology of the disease. According to the cholinergic hypothesis, lost cholinergic neurons affect cholinergic signaling, which plays a crucial role in learning, memory, and cognitive functions [59]. The use of cholinesterase inhibitors aims at increasing the level of acetylcholine in the brain, which induces symptomatic improvement in a patient’s condition [60]. There are indications that Hed and Het may act as cholinesterase inhibitors [61]. Moreover, oxidative stress is believed to be a serious contributor to the progression of Alzheimer’s disease [62]. Prepared systems were characterized by improved inhibitory activity concerning pure compounds. In the case of Hed, the pure compound caused 9.24 ± 0.81% of AChE and 13.86 ± 1.62% of BChE inhibition when systems suppressed AChE and BChE activity by 44.03 ± 4.26% and 56.7 ± 0.95%, respectively, for Hed/HP-β-CD 1:1 and 55.98 ± 1.60% and 67.24 ± 2.87%, respectively, for Hed/HP-β-CD 1:2. When it comes to Het, the unmodified compound inhibited AChE by 6.49 ± 0,79%, whereas BChE was inhibited by 12.96 ± 3.15%. Obtaining amorphous inclusion complexes contributed to increasing the inhibitory activity by 39.91 ± 1.54% and 54.39 ± 2.68% (Het/HP-β-CD 1:1) and by 56.04 ± 2.91% and 77.54 ± 0.90% (Het/HP-β-CD 1:2) with respect to AChE and BChE, respectively. What is more, a correlation was found between increased solubility and the effect on the ability to suppress enzyme activity. Solubility-enhancing complexes can be defined as those giving the best inhibitory activity to cholinesterases. When it comes to antioxidant activity, there is some pattern in which higher solubility translates into a better ability to neutralize the DPPH radical. In conclusion, the results of the in vitro biological studies indicated that the obtained systems improved the activity of the tested compounds, which can be directly correlated with the increased solubility.

## 4. Materials and Methods

### 4.1. Materials

All materials including the tested compounds—hesperidin (Hed, purity > 80%) and hesperetin (Het, purity > 95%)—and 2-hydroxypropyl-β-cyclodextrin (HP-β-CD) were supplied by Sigma-Aldrich (Sigma-Aldrich, St. Louis, MO, USA), except for dimethyl sulfoxide (DMSO), potassium persulfate czda and sodium hydroxide (Avantor Performance Materials Poland S.A., Gliwice, Poland), acetic acid 98–100% and sodium chloride (POCH, Gliwice, Poland), the analytical weighed amount of HCl 1 N (Chempur, Piekary Slaskie, Poland) and sodium dihydrogen phosphate (PanReac AppliChem ITW Reagents, Darmstadt, Germany), and methanol of an HPLC grade (J. T. Baker, Center Valley, PA, USA). High-quality pure water was prepared using a Direct-Q 3 UV purification system (Millipore, Molsheim, France; model Exil SA 67120). Prisma HT, GIT/BBB lipid solution, and acceptor sink buffer were supplied by Pion Inc. (Forest Row, East Sussex, United Kingdom).

### 4.2. Preparation of the Systems

Systems of Hed and Het with HP-β-CD were obtained by the means of solvent evaporation technique in 1:1 and 1:2 molar ratios. To obtain all the complexes, the following procedure was applied: 70 mg of compound was added to a conical flask containing 140 mL of methanol and placed in an ultrasound bath for about 3 min to obtain a lucid solution. Then, an accurately weighted amount of HP-β-CD was added and stirred to a visibly clear solution. After that, the mixture was poured into a round bottom flask and placed in a rotary evaporator (Buchi, Switzerland) to remove methanol under reduced pressure. The water bath was heated up to 40 °C. The process took enough time to visually dry the content of the flask plus 20 min extra to make sure all solvent evaporated. The obtained systems were taken out from the flask with the use of a metal spatula.

Physical mixtures were prepared by weighing an accurate amount of compound and HP-β-CD with respect to the molar ratio and mixing two ingredients in a mortar for 10 min.

The prepared systems were stored in a desiccator at the temperature of 22 °C between studies.

### 4.3. Identification of Obtained Systems

#### 4.3.1. Fourier Transform Infrared Spectroscopy and Density Functional Theory (DFT) Calculations

The FT-IR–ATR spectra of Hed, Het, and HP-β-CD as well as their systems at a molar ratio of 1:1 and 1:2 were collected on an IRTracer-100 spectrophotometer (Shimadzu Corp., Kyoto, Japan). All spectra were measured between 400 and 4000 cm^−1^ in the absorbance mode. The following spectrometer parameters were used: resolution: 4 cm^−1^, number of scans: 400, apodization: Happ–Genzel. The sample was placed directly on the ATR crystal. Solid samples were pressed against the ATR crystal, and the ATR–FT-IR spectrum was scanned. The results were interpreted by comparing the FT-IR peaks of pure samples with those of prepared complexes.

The molecular geometries of Hed and Het were optimized using the Density Functional Theory (DFT) method with Becke’s three-parameter hybrid functional (B3LYP) implemented with the standard 6-311G(d,p) as a basis set. The calculations of normal mode frequencies and intensities were also performed. We applied the PL-Grid platform (website: www.plgrid.pl, accessed on 12 March 2021) equipped with the Gaussian 09 package (Wallingford, CT, USA) for DFT calculation [63]. The GaussView (Wallingford, CT, USA, Version E01) program was used to propose an initial geometry of the investigated molecules and for visual inspection of the normal modes [64]. The experimental results were compared with calculations. Ultimately, the characteristic bands of Hed and Het and assignments are listed in Appendix A, respectively.

#### 4.3.2. X-ray Powder Diffraction

The XRPD patterns of pure Hed and Het and their binary systems with HP-β-CD were recorded at ambient temperature using a Bruker AXS D2 Phaser diffractometer (Bruker, Germany) with CuKα radiation (1.54060 Å). The tube voltage and current were 30 kV and 10 mA, respectively. The samples were scanned from 5° to 40° with a step size of 0.02° and a counting rate of 2 s·step^−1^. The analysis of the acquired data was performed using Origin 2021b software (OriginLab Corporation, Northampton, MA, USA).

#### 4.3.3. Differential Scanning Calorimetry

Thermal analysis was performed using a DSC 214 Polyma differential scanning calorimeter (Netzsch, Selb, Germany). Samples of about 9–10 mg were placed in crimped aluminum pans with a small hole in the lid. The measurements were performed at a constant heating rate of 10° K·min^−1^ under a nitrogen atmosphere with a flow rate of 30 mL·min^−1^. Two heating cycles were applied. The samples were first heated up to 100 °C and held at this temperature for 5 min to remove water, then cooled down to 25 °C and heated again to 275 °C.

### 4.4. Physicochemical Properties

#### 4.4.1. Solubility Studies

Concentrations of Hed and Het during solubility, dissolution rate, and permeability studies were measured by high-performance liquid chromatography with the DAD detector (HPLC-DAD). In this study, a Shimadzu Nexera (Shimadzu Corp., Kyoto, Japan) equipped with SCL-40 system controller; DGU-403 degassing unit; LC-40B XR solvent delivery module; SIL-40C XR auto sampler; CTO-40C column oven; SPD-M40 photo diode array detector, was used. For the stationary phase, a Dr. Maisch ReproSil-Pur Basic-C18 100 Å column, 5 µm particle size, 250 × 4.60 mm (Dr. Maisch, Ammerbuch-Entringen, Germany), was used. The mobile phase was methanol/0.1% acetic acid (65:35 *v*/*v*). The mobile phase was vacuum-filtered through a 0.45 µm nylon filter (Phenomenex, CA, USA). The experimental conditions were as follows: 0.9 mL·min^−1^ flow rate, wavelength 280 nm for Hed and 288 nm for Het, and the column temperature at 30 °C. The injection volume differed depending on the assay. For the solubility study, it was 1 µL, whereas for the dissolution rate and permeability assays, it was 10 µL. The method duration time was 10 min. The retention time was 4.17 min for Hed and 5.83 min for Het.

##### Media for Solubility and Dissolution Rate Studies

A total of 0.1 M HCl was obtained from the analytical weighed amount in accordance with the BASF SE recommendations. Phosphate buffer at pH 6.8 was prepared according to the following description: in a 1000 mL volumetric flask, we placed 250 mL of 0.2 N potassium dihydrogen phosphate solution, then added 112 mL of 0.2 N sodium hydroxide solution and filled the mixture up to 1000 mL with distilled water. High-quality pure water was prepared using a Direct-Q 3 UV purification system (Millipore, Molsheim, France, model Exil SA 67120).

##### Solubility Studies

An excess number of obtained systems and physical mixtures was placed in a 10 mL glass tube; then, 5.0 mL of medium (water, HCl 0.1 N or phosphate buffer (pH 6.8)) was added. All samples were mixed using a vortex mixer for 30 s and left at room temperature for 24 h. The obtained solutions were filtered through a 0.2 μm nylon membrane filter (Sigma-Aldrich, St. Louis, MO, USA) and analyzed for Hed and Het content using the developed and validated the HPLC method. The analysis was performed in triplicate.

#### 4.4.2. Dissolution Studies

The dissolution study was performed in the paddle apparatus. Hed, Het, and their systems with HP-β-CD were weighed to gelatin capsules in the amount corresponding to 15 mg of pure substance, which were later implemented to springs to prevent flotation on the surface of the medium. The study was carried out at a pH of 6.8. The vessels were filled with 500 mL of phosphate buffer, the temperature was maintained at 37 °C, and the paddles were set at the stirring speed of 75 rotations per minute. The 5.0 mL samples were withdrawn at predetermined time points with the replacement of equal volumes of temperature-equilibrated media and filtered through a membrane filter (0.2 μm). The dissolution profiles were compared with the use of two-factor values *f*_1_ and *f*_2_ [65] implemented by Moore and Flanner [66], according to the following equations:$$f_{1}=\frac{\sum_{j=1}^{n}\left|Rj-Tj\right|}{\sum_{j=1}^{n}Rj}\times 100\;f_{2}=50\times \log\left({\left(1+\left(\frac{1}{n}\right)\;\sum_{j=1}^{n}{\left|Rj-Tj\right|}^{2}\right)}^{-\frac{1}{2}}\times 100\right)$$
where *n* is the number of time points, *R_j_* is the percentage of the reference dissolved substance in the medium, *T_j_* is the percentage of the dissolved tested substance, and *t* is the time point. Dissolution profiles are described as similar when the *f*_1_ value is close to 0, or *f*_2_ is close to 100 (between 50 and 100) [66]. The results of fitting factors *f*_1_ and *f*_2_ are presented in Appendix A.

#### 4.4.3. Permeability Studies

In vitro gastrointestinal (GIT) and blood–brain barrier (BBB) permeability was studied using the PAMPA (Parallel Artificial Membrane Permeability Assay) model. The sandwich consists of two 96-well microfilter plates. The PAMPA system contains two chambers: the donor at the bottom and the acceptor chamber at the top. The chambers are separated by a 120 μm thick microfilter disc coated with a 20% (*w*/*v*) dodecane solution of a lecithin mixture (Pion, Inc.). The analyzed samples were prepared by dissolving an excess amount of pure compound and systems in water. In addition, the pure compound was dissolved in DMSO. The donor solution was adjusted to pH ≈ 6.8 for GIT application and to pH ≈ 7.4 for BBB application using 0.5 M NaOH. The plates were combined and then incubated for 4 h for both models in a humidity-saturated atmosphere with the temperature set at 37 °C. The apparent permeability coefficient (*P_app_*) was calculated according to the following equation:$$P_{app}=\frac{-ln\left(1-\frac{C_{A}}{C_{equilibrium}}\right)}{S\times \left(\frac{1}{V_{D}}+\frac{1}{V_{A}}\right)\times t}$$
where *VD* is donor volume; *VA* is acceptor volume; *Cequilibrium* is equilibrium concentration, $C_{equilibrium}=\frac{C_{D}\times V_{D}+C_{A}\times V_{A}}{V_{D}+V_{A}}$; *S* is membrane area; and *t* is incubation time (in seconds). Compounds with the value of *P_app_* in GIT model below 0.1 × 10^−6^ cm s^−1^ are described as low permeable, substances found as medium permeable have a 0.1 × 10^−6 ^cm·s^−1^ ≤ *P_app_* < 1 × 10^−6^ cm·s^−1^, and compounds with a *P_app_*  ≥  1 × 10^−6^ cm·s^−1^ are defined as ones with high permeability [67].

Substances whose *P_app_* in BBB model is <2.0 × 10^−6 ^cm·s^−1^ are defined as low permeable. API with the *P_app_* value in the range of 2.0–4.0 × 10^−6 ^cm·s^−1^ are described as substances with questionable permeability. Compounds with high permeability have the *P_app_* value at the level >4.0 × 10^−6 ^cm·s^−1^ [68].

### 4.5. Biological Assays

#### 4.5.1. Antioxidant Activity Assay

The antioxidant activity was evaluated using the DPPH radical reaction. The investigation of antioxidant properties was performed spectrophotometrically, according to an outlined procedure [69]. A methanol solution of DPPH (0.2 mM) was prepared. The samples were prepared by dissolving an excess amount of system in water. A total of 25.0 µL of DPPH solution was mixed with 175.0 µL of studied solutions in a 96-well plate, and then the plate was shaken and incubated in darkness for half an hour at room temperature. Subsequently, the absorbance (A) was measured at the wavelength set at 517 nm against the blank (the mixture of DPPH solution and methanol). The inhibition of the DPPH radical by the studied samples was calculated via the following equation:$$Inhibition\;ability\;\left(\%\right)=\frac{A-A_{i}}{A_{0}}\times 100\%$$
where *A*_o_ is the absorbance of the control sample, and *A_i_* is the absorbance of studied samples. The analysis was performed in the sextet.

#### 4.5.2. Determination of Acetylcholinesterase (AChE) and Butyrylcholinesterase (BChE) Inhibition

To assess the activity of Hed, Het, and systems, the spectrometric method according to the outlined procedure was applied [70]. A Multiskan GO 1510 (Thermo Fisher Scientific, Vantaa, Finland) plate reader was used for measurements of 96-well plates of the maximum volume of 300 μL. The hydrolysis of acetylthiocholine/butyrylthiocholine caused a color change. The absorbance of the enzymes was measured at a wavelength of 412 nm, 10 min after pipetting on a microplate. The reaction mixture containing 0.1 mL of 0.3 mM 5,5-dithio-bis-(2-nitrobenzoic acid) (DTNB, Sigma Aldrich, Germany), 10 mM NaCl and 2 mM MgCl_2_·6H_2_O solution, 0.575 mL 50 mM Tris-HCl buffer (pH = 8.0), 25 µL of 0.28 units/mL AChE/BChE (Sigma Aldrich, Germany), and 0.2 mL of the tested system was measured at a wavelength of 405 nm and at a temperature of 22 °C. The measurement was conducted after 20 min (BChE) or 60 min (AChE) after adding all ingredients into a microplate. The blank sample contained Tris-HCl buffer instead of the tested compounds. All samples were analyzed in six independent replicates. The inhibitory activity of each enzyme was calculated with the use of following equation:$$Inhibition\;ability\;\left(\%\right)=100-\frac{Ap-Abp\times 100}{Ak-Ab}$$
where *Ak* is absorbance of control, *Ab* is the blank of the control, *Ap* is the absorbance of the sample, and *Abp* is the blank of the sample.

### 4.6. Docking

Hed and Het molecules were drawn manually by using the Avogadro 1.1.1 software [71] and optimized within the UFF force field [72] (5000 steps, steepest descent algorithm). The β-CD and 2HP-β-CD molecules were prepared, relying on the available crystal structures of β-CD and manually substituted by the 2HP groups, according to the two alternative substitution patterns by using Avogadro 1.1.1. Upon this modification, the two 2HP-β-CD structures were optimized within the UFF force field. From the known 2HP-β-CD molar mass, it was deduced that the most probable number of the 2HP moieties present in one molecule was equal to 6; however, the exact substitution pattern remained unknown. Thus, we performed calculations for the two alternative compounds: (1) 2HP-β-CD* containing all 2HP groups substituted to the O_(2)_ hydroxyl oxygen atoms of β-CD (atom numbering in accordance with the IUPAC recommendations), and (2) 2HP-β-CD^#^ containing all 2HP groups substituted to the O_(6)_ hydroxyl oxygen atoms of the hydroxymethyl groups present in β-CD. These two cases represent the two topologically limiting substitution patterns, placing all 2HP moieties at opposite sides of the β-CD torus. The guest–host docking was carried out by using the AutoDock Vina software [73]. The procedure of docking was performed within the cuboid region that covered the whole 2HP-β-CD molecule. All the default procedures and algorithms implemented in AutoDock Vina were applied during docking. In addition to the flexibility of the guest molecules, the rotation of the 2HP and hydroxymethyl groups was allowed as well.

### 4.7. Molecular Dynamics

The molecular dynamics (MD) simulations concerned all possible six guest–host complexes. The initial structures of such complexes were based on the most energetically favorable poses identified during the docking study. The MD simulations were carried out within the GROMACS 2016.4 package [74]. To describe the interactions within the system, the CHARMM [75,76] force field was applied and the CHARMM-GUI online server [77] was used to generate the parameters. The considered MD systems consisted of one complex solvated by water molecules (TIP3P model [78]) within a cubic computational box simulated under periodic boundary conditions. The box edges (of initial dimensions corresponding to 5.5 × 5.5 × 5.5 nm^3^) were preoptimized by a 2 ns constant-pressure MD equilibration at 1 bar and 298 K, ensuring an effective solvent density in the subsequent production simulations. After equilibration, all simulations were carried out for 100 ns and the trajectory was saved every 1 ps. The temperature was maintained close to its reference value (298 K) by applying the V-rescale thermostat [79], whereas for the constant pressure (1 bar, isotropic coordinate scaling), the Parrinello–Rahman barostat [80] was used with a relaxation time of 0.4 ps. The equations of motion were integrated with a time step of 2 fs using the leap-frog scheme [81]. The translational center-of-mass motion was removed every timestep separately for the solute and the solvent. The full rigidity of the water molecules was enforced by application of the SETTLE procedure [82]. The hydrogen-containing solute bond lengths were constrained by application of the LINCS procedure with a relative geometric tolerance of 10^−4^ [83]. The electrostatic interactions were modeled by using the particle-mesh Ewald method [84] with a cut-off set to 1.2 nm, while Van der Waals interactions (Lennard–Jones potentials) were switched off between 1.0 and 1.2 nm.

## 5. Conclusions

The obtained amorphous inclusion complexes in the current study seem to be a promising delivery system to boost the overall pro-healthy activity of hesperidin and its aglycone—hesperetin. The presence of hesperidin and hesperetin with HP-β-CD interactions, confirmed by FT-IR, XRPD, and DSC techniques, resulted in a significant improvement in the solubility of both polyphenols—over 1000 and 2000 times, respectively. As an effect of interactions in the case of hesperidin, the improvement of solubility that correlated with better permeability in the model simulating the walls of the gastrointestinal system was observed. Moreover, the improved solubility of the hesperidin and hesperetin increased their antioxidant potential as well as the ability to inhibit cholinesterases. References [85,86,87] are cited in the Appendix A.

## Figures and Tables

**Figure 1 ijms-23-04000-f001:**
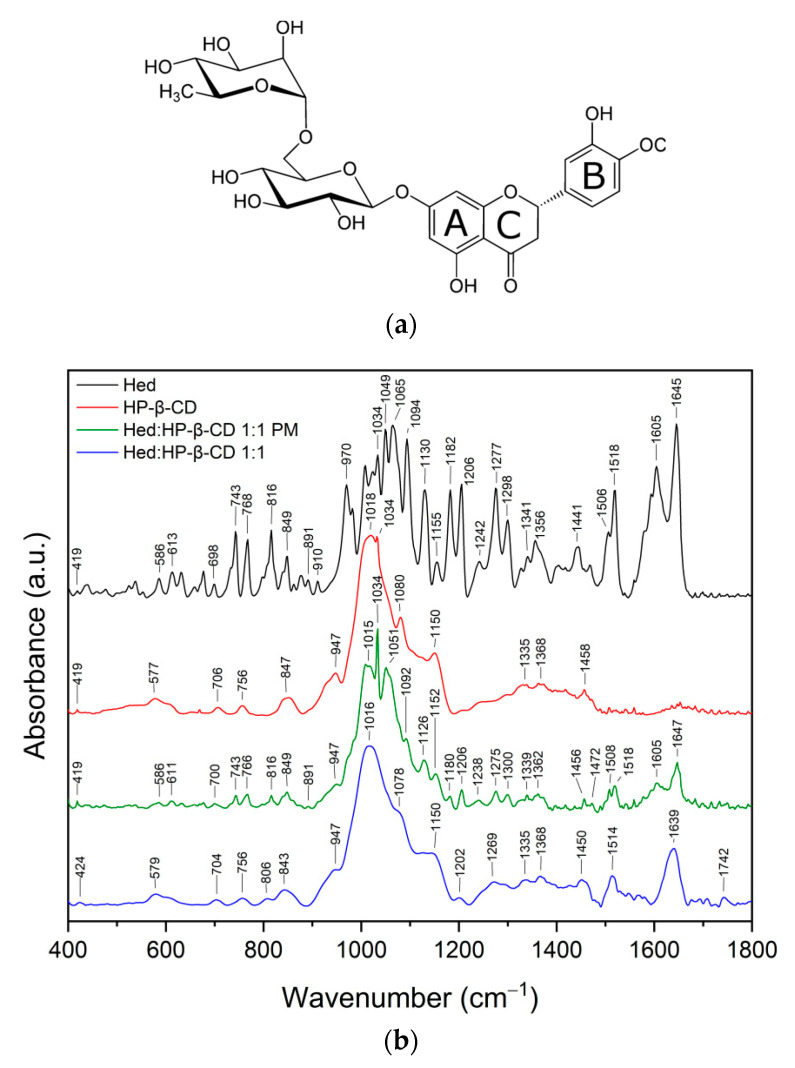
Structure of Hed (**a**). The results of FT-IR analysis of Hed (black), HP-β-CD (red), Hed/HP-β-CD 1:1 physical mixture (green), Hed/HP-β-CD 1:1 physical mixture (blue) (**b**), and Hed (black), HP-β-CD (red), Hed/HP-β-CD 1:2 physical mixture (green), Hed/HP-β-CD 1:2 physical mixture (blue) (**c**); range 400–1800 cm^−1^.

**Figure 2 ijms-23-04000-f002:**
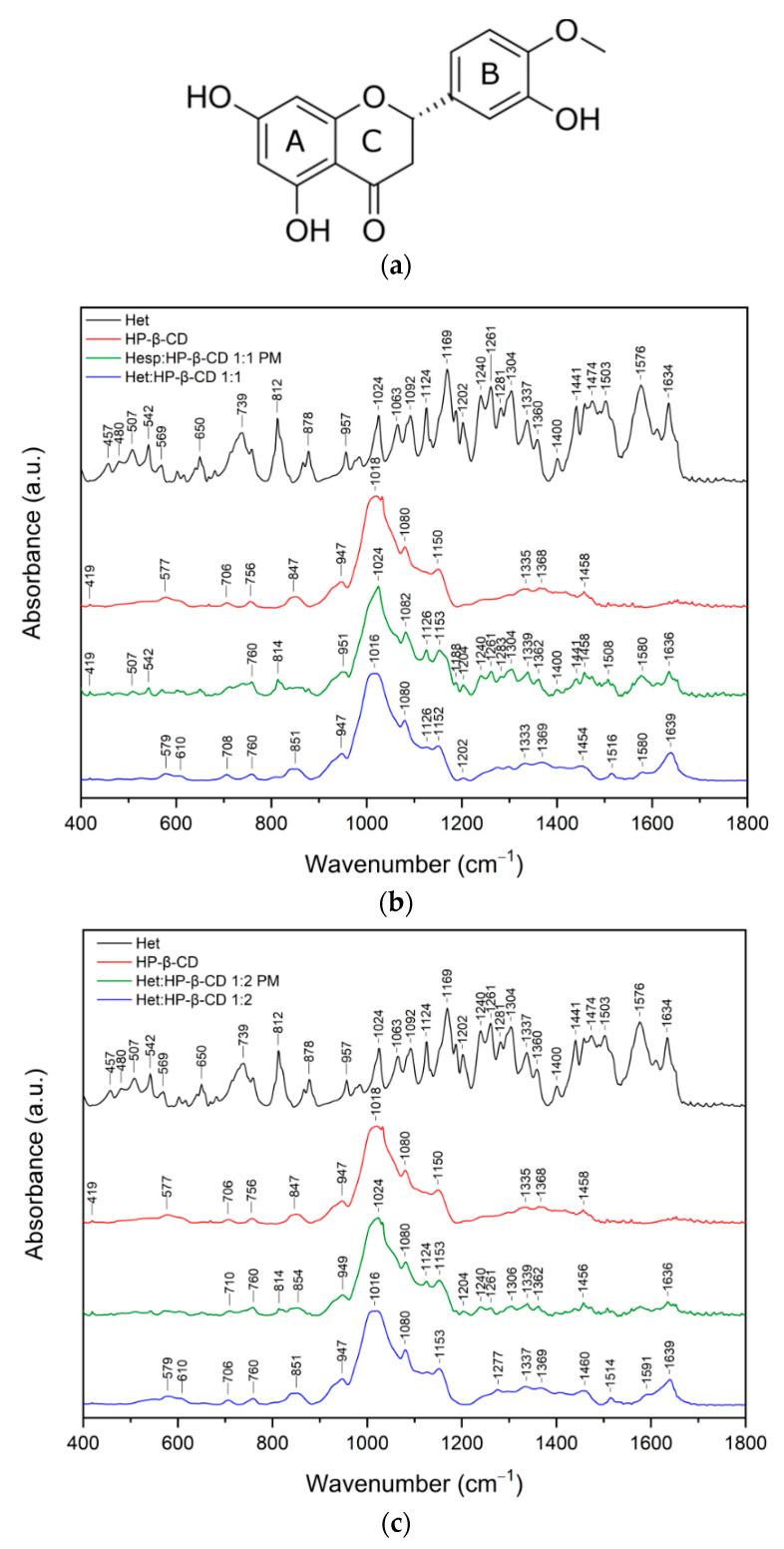
Structure of Het (**a**). The results of FT-IR analysis of Het (black), HP-β-CD (red), Het/HP-β-CD 1:1 physical mixture (green), Het/HP-β-CD 1:1 physical mixture (blue) (**b**), and Het (black), HP-β-CD (red), Het/HP-β-CD 1:2 physical mixture (green), Het/HP-β-CD 1:2 physical mixture (blue) (**c**); range 400–1800 cm^−1^.

**Figure 3 ijms-23-04000-f003:**
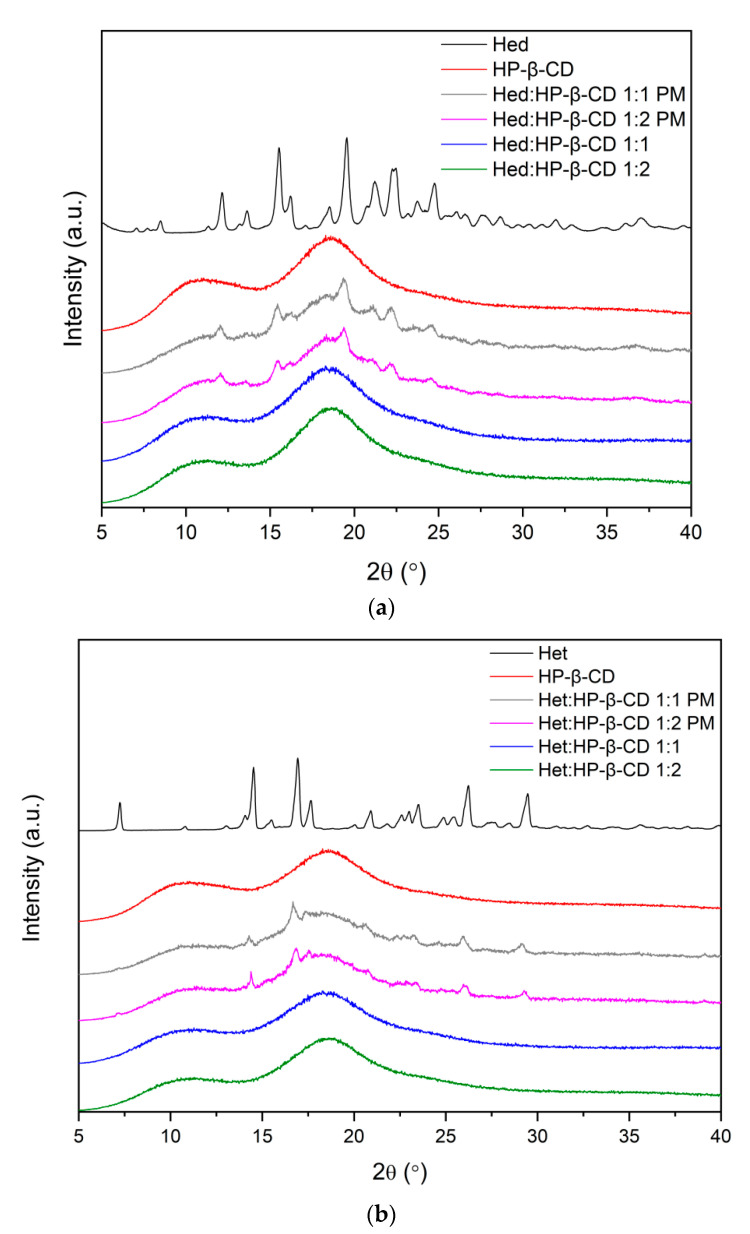
The results of XRPD analysis of Hed (**a**) and Het (**b**) physical mixtures and systems.

**Figure 4 ijms-23-04000-f004:**
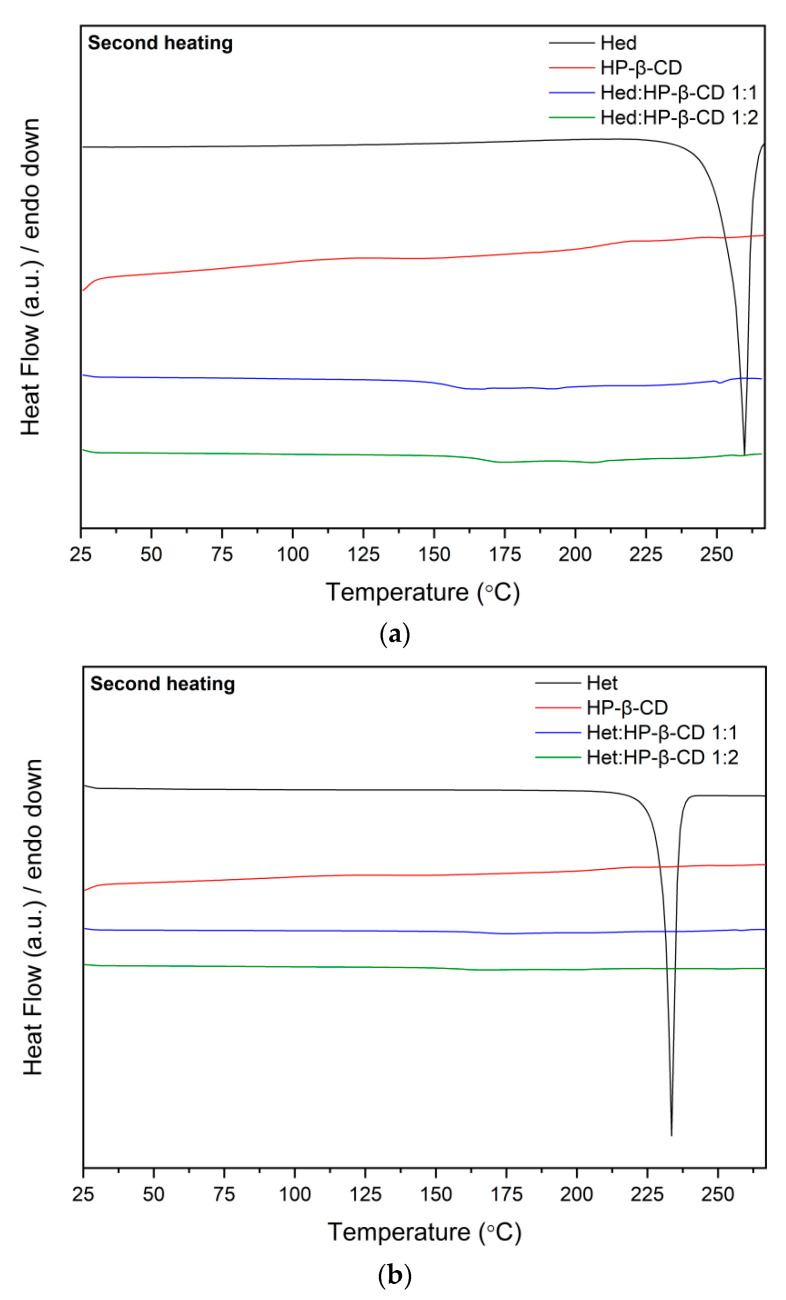
The results of DSC analysis from the second heating scan of Hed (**a**) and Het (**b**) systems.

**Figure 5 ijms-23-04000-f005:**
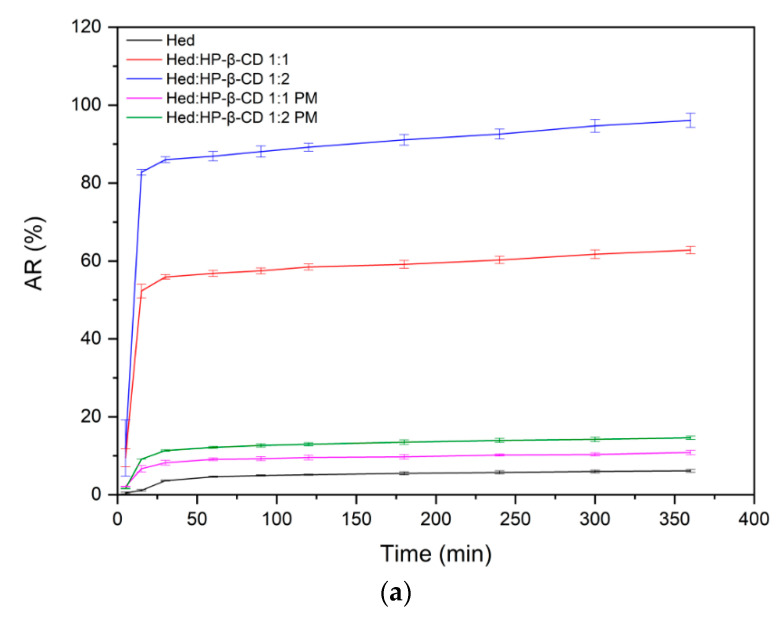
The results of dissolution rate studies of Hed (**a**) and Het (**b**) systems and physical mixtures.

**Figure 6 ijms-23-04000-f006:**
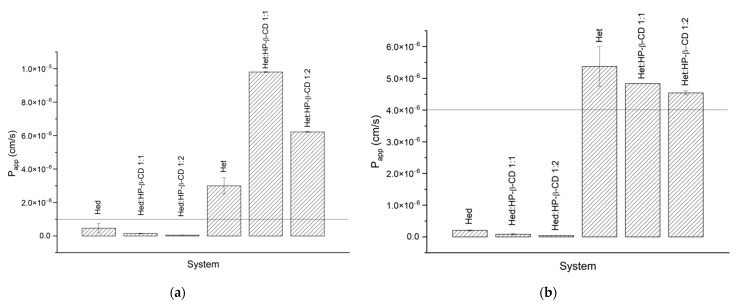
The results of PAMPA (**a**) GIT and (**b**) BBB assays.

**Figure 7 ijms-23-04000-f007:**
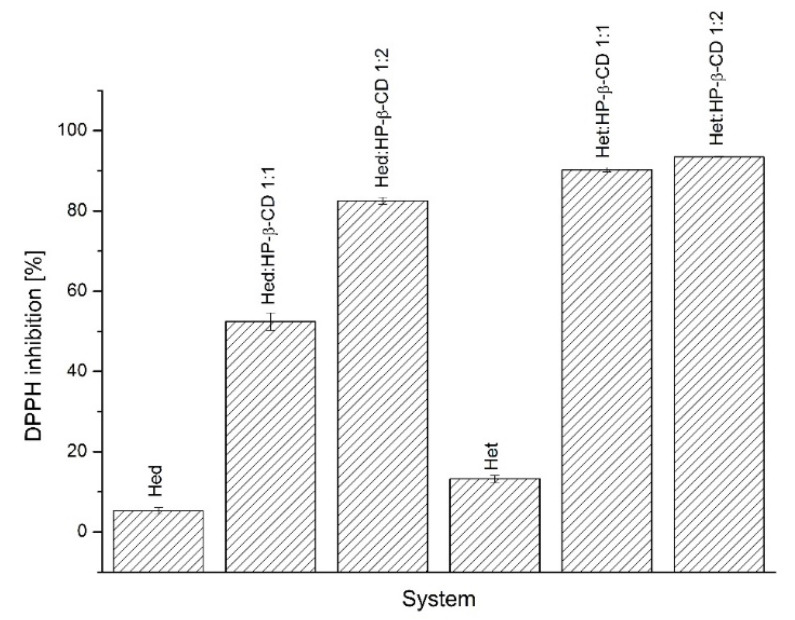
The results of the DPPH antioxidant assay of Hed and Het systems with HP-β-CD.

**Figure 8 ijms-23-04000-f008:**
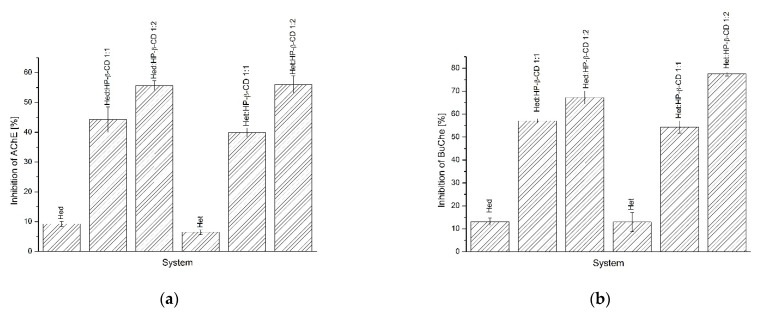
The results of AChE (**a**) and BChE (**b**) inhibition assays of Hed and Het systems with HP-β-CD.

**Figure 9 ijms-23-04000-f009:**
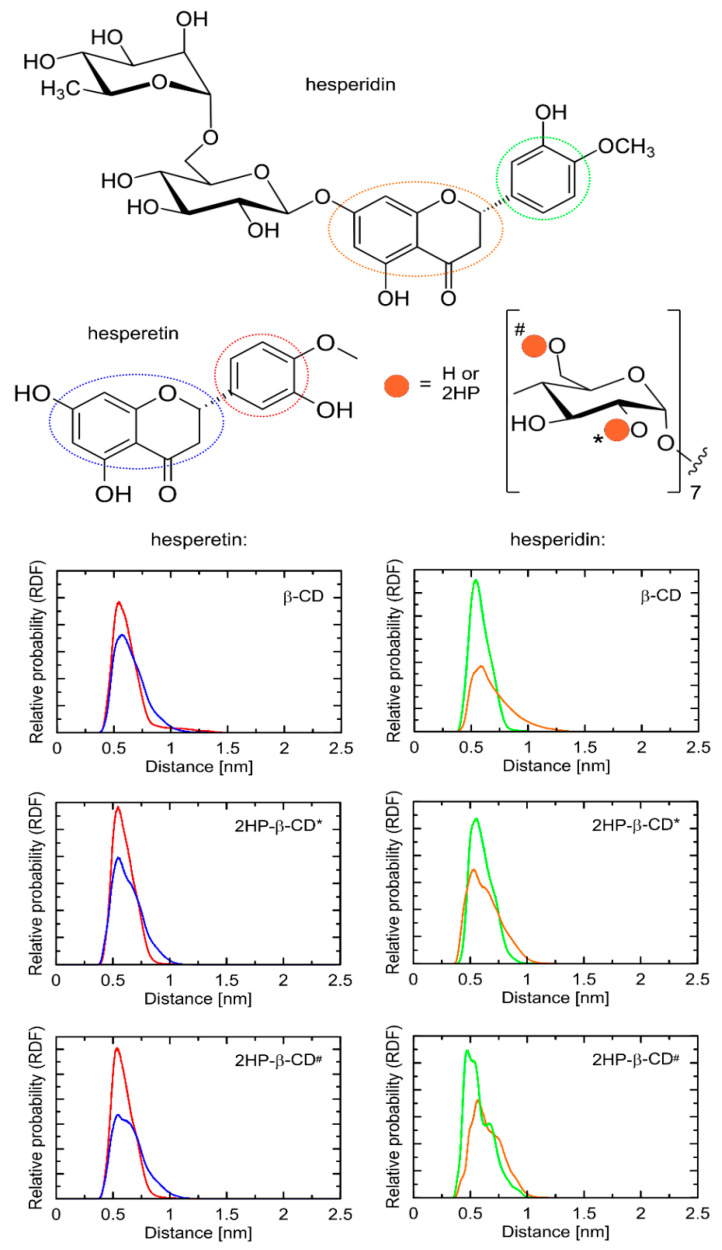
Radial distribution functions (RDFs) calculated for distances between selected fragments of the studied systems. Only the “anchoring site” of guest molecules was considered and split into two parts. In the case of hosts, the inner channel (aliphatic carbon atoms attached to glycosidic oxygen atoms) were the reference for calculating RDFs. The color code is maintained throughout the figure. 2HP-β-CD* denotes that all 2HP groups are substituted to the O(2) hydroxyl oxygen atoms of β-CD; 2HP-β-CD# denotes that all 2HP groups are substituted to the O(6) hydroxyl oxygen atoms of the hydroxymethyl groups present in β-CD.

**Figure 10 ijms-23-04000-f010:**
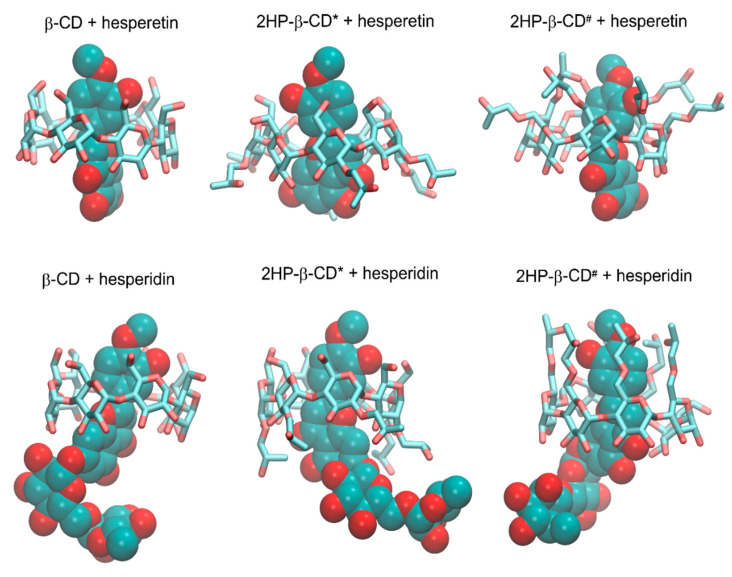
The exemplary structures of the guest–host complexes, generated during MD simulations. The two alternative substitution patterns of β-CD by the 2HP groups were considered. Hydrogen atoms are not shown. 2HP-β-CD* denotes that all 2HP groups are substituted to the O(2) hydroxyl oxygen atoms of β-CD; 2HP-β-CD# denotes that all 2HP groups are substituted to the O(6) hydroxyl oxygen atoms of the hydroxymethyl groups present in β-CD.

**Table 1 ijms-23-04000-t001:** The results of solubility studies of Hed and Het systems and physical mixtures. ↑ means Multiple relationships compared to Concentrations of Hed or Het.

System	Concentration (mg·mL^−1^) (↑-Improvement of Solubility (Fold))		
	Water	Phosphate Buffer pH 6.8	HCl 0.1 N
Hed	0.009 ± 0.001	0.009 ± 0.002	0.008 ± 0.003
Hed/HP-β-CD 1:1	1.709 ± 0.012 (↑190)	1.861 ± 0.023 (↑207)	1.663 ± 0.004 (↑208)
Hed/HP-β-CD 1:2	9.280 ± 0.011 (↑1031)	10.588 ± 0.073 (↑1176)	9.000 ± 0.043 (↑1125)
Hed/HP-β-CD 1:1 PM	0.439 ± 0.003 (↑49)	0.432 ± 0.001 (↑48)	0.396 ± 0.0003 (↑50)
Hed/HP-β-CD 1:2 PM	0.679 ± 0.001 (↑75)	0.692 ± 0.001 (↑77)	0.605 ± 0.0005 (↑76)
Het	0.008 ± 0.0001	0.015 ± 0.0002	0.006 ± 0.0001
Het/HP-β-CD 1:1	6.965 ± 0.021 (↑871)	7.018 ± 0.042 (↑468)	6.238 ± 0.015 (↑1040)
Het/HP-β-CD 1:2	12.546 ± 0.091 (↑1568)	12.756 ± 0.097 (↑850)	12.933 ± 0.028 (↑2156)
Het/HP-β-CD 1:1 PM	1.401 ± 0.002 (↑175)	2.401 ± 0.003 (↑160)	1.642 ± 0.001 (↑274)
Het/HP-β-CD 1:2 PM	2.826 ± 0.001 (↑353)	4.969 ± 0.002 (↑331)	3.750 ± 0.004 (↑625)

**Table 2 ijms-23-04000-t002:** The average nonbonded interactions between guest and host molecules calculated on the basis of the MD trajectory and docking study (last row). The changes of the solvent-accessible surface area (SASA) parameter occurring upon complexation are given as well. In the case of docking data, the range of energies corresponding to analogous docking poses is given. 2HP-β-CD* denotes that all 2HP groups are substituted to the O(2) hydroxyl oxygen atoms of β-CD; 2HP-β-CD# denotes that all 2HP groups are substituted to the O(6) hydroxyl oxygen atoms of the hydroxymethyl groups present in β-CD.

Guest–Host Interactions	Hesperidin + β-CD	Hesperidin + 2HP-β-CD*	Hesperidin + 2HP-β-CD^#^	Hesperetin + β-CD	Hesperetin + 2HP-β-CD*	Hesperetin + 2HP-β-CD^#^
Lennard–Jones (kJ/mol)	−100.2	−117.6	−129.6	−92.7	−102.4	−106.3
Coulombic (kJ/mol)	−25.8	−40.2	−54.3	−14.1	−24.6	−15.1
Hydrogen bonding (occurrence/timeframe)	0.653	1.366	2.079	0.194	1.024	0.592
SASA (nm^2^)	−4.18	−6.04	−6.52	−4.13	−5.17	−4.47
Binding energy (kJ/mol)	−23.4–−20.5	−24.7–−20.5	−28.5–−25.5	−26.4–−24.3	−28.0–−26.4	−30.5–−29.3

## Data Availability

Data are available in a publicly accessible repository.

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
