# Peer review of "Amorphous Inclusion Complexes: Molecular Interactions of Hesperidin and Hesperetin with HP-Β-CD and Their Biological Effects"

_ijms, 2022, doi:10.3390/ijms23074000_

Round 1

Reviewer 1 Report

The paper entitled: “ Amorphous Inclusion Complexes:  Molecular Interactions of Hesperidin and Hesperitin with HP-bCD and their biological Effects” described the preparation of solid dispersions of Hesperidin and Hesperitin with HP-bCD by solvent evaporation method. FTIR spectroscopy, X-ray diffraction analysis, DSC studies  as well as the in silico and molecular dynamics studies lead to characterize the inclusion complexes obtained. Solubility behavior and drug release together with permeability studies are included.

In general, the study is well-designed and written but some important considerations must be taken into account in order to improve the quality of the study s describe above:

X-ray diffraction analysis

  1. The physical mixtures patterns are not shown in this study . It is well known that the physical mixtures of components allows to determine the inclusion behavior in the solid dispersion by comparing the X-ray pattern  of both systems.

DSC studies

  1. Why have different thermal conditions been carried out? Two cycles (heating-cooling-heating) for HP-bCD whereas only one cycle of heating (25-250°C) in polyphenols-HPbCD systems. The same conditions should be applied, only one heating cycle in both cases.

Solubiliy studies

  1. Authors should explain in more detail how pH conditions affects to solubility increase of Hesperidin and Hesperitin in presence of HPbCD.

Solubility diagrams (Higuchi diagrams: AL or BS ) can be performed in order to calculate the stability constants  of both complexes.

Dissolution rate studies

  1. In order to complete the dissolution study of polyphenols-HPbCD systems under physiological conditions, a preliminary release study under gastric conditions (pH 1.2 for 1.5 h.) must be tested followed by the polyphenols release in pH 6.8.

Conclusions

  1. The conclusions should be rewritten in more detail.

Author Response

The Authors provide a point-by-point response to the reviewer’s comments and upload it as a PDF file. 

Reviewer 2 Report

The submitted manuscript concerns research on obtaining the Hesperidin (Hed) and Hesperetin (Het) systems from HP-β-CD by solvent evaporation. Hed and Het are flavanones that can be occur in citrus fruits (sweet oranges, clementines, mandarins, lime, lemon, grapefruit). These compounds have a number of pharmacological activities, i.e. antitumor, antidiabetic and antimicrobial. Therefore, their application can be diverse. The authors have successfully undertaken to obtain the Het and Hed systems with higher pharmacological properties, e.g. solubility, dissolution rate, and thus also antioxidant properties. This knowledge can be valuable when these flavonones are used as additives in food products as they are much easier to supply in this form. I accept the manuscript as presented.

Author Response

The Authors provide response to the reviewer’s comments and upload it as a PDF file. 

Round 2

Reviewer 1 Report

Despite what the authors stated, I still think that a previous study
on the compounds release in gastric medium should have been carried
out althoug the fact that absorption occurs mainly in the intestinal
medium. The influence of an acidic pH prior
to release in an alkaline medium
must be performed since it is
appropriate way to approach the release studies